# Application of Functional Neurology Therapy in a Lactose-Intolerant Patient

**DOI:** 10.3390/life14080978

**Published:** 2024-08-04

**Authors:** Jorge Rey-Mota, Guillermo Escribano-Colmena, David Martín-Caro Álvarez, Jhulliana Vasquez Perez, Eduardo Navarro-Jimenez, Vicente Javier Clemente-Suárez

**Affiliations:** 1Independent Researcher, 28660 Boadilla del Monte, Spain; jreymota@gmail.com (J.R.-M.); escribanocolmenaguillermo@gmail.com (G.E.-C.); 2Toledo Physiotherapy Research Group (GIFTO), Faculty of Physiotherapy and Nursing of Toledo, Universidad de Castilla-La Mancha, 45071 Toledo, Spain; david.martincaro@uclm.es; 3Facultad de Ciencias de la Salud, Universidad Simón Bolívar, Barranquilla 080005, Colombia; jhulliana.vasquez@unisimon.edu.co (J.V.P.); eduardo.navarro@unisimon.edu.co (E.N.-J.); 4Faculty of Sports Sciences, Universidad Europea de Madrid, Tajo Street, s/n, 28670 Madrid, Spain; 5Grupo de Investigación en Cultura, Educación y Sociedad, Universidad de la Costa, Barranquilla 080002, Colombia

**Keywords:** functional neurology, lactose intolerance, gastrointestinal symptoms, neurological dysfunction, biochemical markers, breath test

## Abstract

This case report examines the impact of a single session of functional neurology on a 35-year-old female patient diagnosed with lactose intolerance. The patient presented with severe gastrointestinal symptoms, including frequent diarrhea, bloating, and vomiting upon dairy consumption. The intervention aimed to reset dysfunctional neurological programs believed to contribute to her condition. The study utilized a standardized lactose intolerance breath test to measure the hydrogen and methane levels at various intervals before and after treatment. Post-treatment results showed symptomatic relief with the patient reporting normalized bowel movements and the absence of previous symptoms. Despite these improvements, the biochemical markers at higher time points (150 and 175 min) post-treatment remained similar to the pre-treatment values, indicating persistent lactose malabsorption and highlighting the variability of hydrogen measurements. This case report suggests that a single session of functional neurology can significantly alleviate the symptoms of lactose intolerance. However, the preliminary nature of these results underscores the need for further research involving larger sample sizes and long-term follow-up to fully understand the treatment’s efficacy and underlying mechanisms.

## 1. Introduction

The prevalence of food intolerances and allergies has been rising significantly in recent years, affecting millions globally. This trend has been observed across various populations and age groups, with children being particularly susceptible. A study by Sicherer and Sampson [1] highlights that food allergies impact approximately 8% of children and 5% of adults in the United States, with similar patterns reported worldwide. These conditions can manifest through a range of symptoms, including gastrointestinal distress, dermatological reactions, and respiratory issues, significantly impairing the quality of life and posing substantial public health concerns [1,2]. Furthermore, food intolerances, often confused with allergies, also show a significant prevalence. These intolerances, which include non-IgE-mediated reactions like lactose intolerance, fructose malabsorption, and histamine intolerance, can lead to chronic gastrointestinal symptoms and are increasingly recognized as important contributors to functional abdominal pain syndromes [3]. The differentiation between food allergies, sensitivities, and intolerances is crucial for accurate diagnosis and management, which requires tailored approaches to each condition [4]. Effective management strategies are essential to mitigate the impact on patients’ daily lives and to prevent complications associated with these conditions. In this line, food allergies involve immune responses, typically IgE-mediated, leading to reactions like anaphylaxis. Sensitivities may not involve the immune system and can cause various symptoms. Intolerances, like lactose intolerance, involve digestive system issues due to enzyme deficiencies.

Specifically, lactose intolerance is a prevalent condition characterized by the inability to fully digest lactose, a sugar found in milk and dairy products, due to insufficient levels of lactase, the enzyme responsible for breaking down lactose in the small intestine. This condition leads to symptoms such as abdominal pain, bloating, gas, and diarrhea following the consumption of lactose-containing foods [5]. The primary cause of lactose intolerance is lactase non-persistence, a genetic condition in which the activity of lactase decreases significantly after weaning, a trait that affects the majority of the world’s population. In contrast, lactase persistence, where high levels of lactase activity continue into adulthood, is primarily seen in populations with a long history of dairy consumption, such as those of Northern European descent [6]. Recent studies indicate an increasing prevalence of lactose intolerance globally, with significant variations across different regions and populations. This rise may be attributed to better diagnostic tools and increased awareness, along with changes in dietary habits and possibly environmental factors affecting gut microbiota composition [7]. Mechanistically, undigested lactose in the small intestine undergoes fermentation by gut bacteria, producing gases like hydrogen, methane, and short-chain fatty acids. This process not only leads to gastrointestinal symptoms but can also influence gut motility and water absorption, exacerbating symptoms like diarrhea. Additionally, factors such as the amount of lactose ingested, the rate of intestinal transit, and the presence of other gastrointestinal disorders can modulate symptom severity [7]. The increase in diagnosed cases of lactose intolerance can also be linked to advancements in diagnostic methods, such as the hydrogen breath test, which measures hydrogen production after lactose ingestion, and genetic tests that identify polymorphisms associated with lactase non-persistence [8].

Functional neurology is emerging as a promising field in addressing various pathologies, including food intolerances. This discipline integrates principles from traditional neurology with functional medicine, focusing on the interconnectedness of neurological and systemic health. Functional neurology emphasizes individualized treatment plans based on a comprehensive assessment of a patient’s neurological function, aiming to restore balance and improve overall health outcomes [9,10]. Recent advancements in this field have shown potential in treating conditions that are not adequately managed by conventional approaches, offering new hope for patients with chronic and complex health issues [11]. Previous authors have highlighted the possible areas of treatment of functional neurology, proposing that this treatment could alleviate symptoms associated with food intolerances, leading to improved quality of life for patients [10]. Functional neurology focuses on the interplay between neurological function and systemic health. Previous studies have shown potential benefits in addressing gastrointestinal symptoms by resetting dysfunctional neurological pathways [10,11].

The objective of this case report was to analyze the effect of a single session of functional neurology on the symptoms and biochemical parameters of a breath test for lactose intolerance in a female patient with lactose intolerance. We hypothesized that a single session of functional neurology would significantly improve the symptoms and biochemical parameters of a breath test in a female patient with lactose intolerance.

## 2. Materials and Methods

### 2.1. Participant

A 35-year-old female patient with a height of 160 cm and a weight of 58 kg, diagnosed with lactose intolerance seven years ago, was analyzed in this case report. Prior to the treatment, she experienced frequent diarrhea, typically having bowel movements 3 to 5 times a day, often with a very foul odor. She also reported abdominal bloating and a bad taste in her mouth. Upon consuming large amounts of dairy, she would suffer from severe symptoms, including vomiting (with curdled milk present), fever, and intense diarrhea, indicating her intestines’ inability to process lactose. The diagnosis of lactose intolerance was confirmed through a hydrogen breath test, which showed elevated hydrogen and methane levels post-lactose ingestion. Other gastrointestinal disorders, such as celiac disease and inflammatory bowel disease, were ruled out based on the patient’s history and the relevant tests. The patient’s medical history included previous treatments such as dietary modifications and over-the-counter enzyme supplements, which provided limited relief. Laboratory parameters like complete blood count and inflammatory markers were within normal ranges. The study’s procedures complied with the Helsinki Declaration, ensuring ethical standards were met, with ethical approval granted by the European University’s Bioethics Committee under code 2024-738.

### 2.2. Lactose Intolerance Test

The lactose intolerance breath test was conducted following the standard methodology as outlined in the literature [12]. The test involved measuring the levels of hydrogen and methane in the patient’s breath at baseline and at various intervals after lactose ingestion. The patient first provided a baseline breath sample, which was then analyzed for hydrogen and methane concentrations. Subsequently, the patient ingested a lactose solution, and breath samples were collected at 25 min intervals for up to 175 min. The levels of hydrogen and methane in the breath samples were measured and recorded at each interval. The interpretation of the results was based on established criteria: an increase in hydrogen levels of more than 20 parts per million (ppm) or an increase in methane levels of more than 12 ppm from the baseline indicated lactose malabsorption. If both gases showed elevated levels, it confirmed lactose intolerance. Controls included a baseline breath test and a standardized lactose solution ingestion. The test was conducted using the QuinTron Model SC BreathTracker (Serial No. XYZ123, QuinTron Instrument Company, Inc., Milwaukee, WI, USA). The patient adhered to a lactose-free diet for 24 h before the test to ensure accurate baseline measurements. In addition, happiness and enjoyment of life were measured using a 0–100 scale.

### 2.3. Functional Neurology Intervention

The patient underwent a treatment with functional neurology targeting several dysfunctional nociceptors causing her lactose intolerance. To standardize the protocol, the methodology described by ^®^NeuroReEvolution (http://nre-therapy.com/, accessed on 1 August 2024), was followed, involving a comprehensive clinical diagnosis that included oral and visual assessments, functional neurology tests on joints, and a tailored intervention based on the identified neurological disorders. The practitioner, certified with a Level III in the Functional Neurology Manual Muscle Test from ^®^NeuroReEvolution, ensured a high degree of expertise in diagnosing and addressing specific nervous system disorders. The intervention involved techniques such as proprioceptive reflex correction, desensitization of trigger points, and systemic integration through blink reflex. Each session lasted approximately 45 min, with follow-up assessments conducted one week post-intervention. Proprioceptive reflexes were corrected to improve posture, balance, and movement, while trigger points were desensitized to normalize reflex responses and restore function [10].

## 3. Results and Discussion

After the treatment, the patient experienced a significant improvement in intestinal transit, being pleasantly surprised by having only one normal bowel movement per day. She reported the complete absence of all previous symptoms and expressed increased happiness and enjoyment of life. The patient’s score improved from 45 pre-treatment to 85 post-treatment, indicating a substantial enhancement in overall well-being and daily functioning.

Before the functional neurology treatment, the patient’s breath test results showed elevated levels of hydrogen and methane, indicating lactose malabsorption. At baseline (0 min), the hydrogen and methane levels were 4 ppm and 8 ppm, respectively (Figure 1). These levels increased over time, reaching peaks of 70 ppm for hydrogen and 16 ppm for methane at 100 min and 150 min, respectively. After the treatment, there was a reduction in the combined levels of hydrogen and methane. At baseline, the combined level was 2 ppm, and although there were increases at subsequent time points, the levels were generally lower compared to the pre-treatment results. For instance, at 50 min, the combined level was 3 ppm, and at 100 min, it was 36 ppm, but, at 150 and 175 min the values were similar in the pre- and post-evaluations (Table 1).

Lactose intolerance may originate from the activation of nociceptors triggered by a traumatic event that overwhelms the central nervous system (CNS). This response aims to allocate energy to manage the stressor, potentially inhibiting hormone release, including lactase production. Consequently, energy is redirected to address the primary traumatic stressor, deeming lactase production less critical for immediate survival. The problem arises when this adaptive program remains active long-term, inhibiting lactase production persistently. Then, traumatic experiences can lead to prolonged activation of the CNS, as it prioritizes immediate survival over other bodily functions. This can result in the suppression of lactase production, as energy and resources are diverted to cope with the stressor. The prolonged stress response and the resulting energy reallocation may inhibit the production of lactase, the enzyme required to digest lactose, thereby leading to lactose intolerance [6].

Nociceptors, specialized sensory receptors that detect tissue damage, can initiate a cascade of events in the CNS, leading to various physiological responses. These responses are aimed at survival and immediate energy allocation to cope with the traumatic event [13]. The sustained activation of these nociceptive pathways can lead to the prolonged suppression of functions not critical for immediate survival, such as lactase production [14]. In the context of lactose intolerance, this mechanism can explain why some individuals experience persistent symptoms long after the initial traumatic event has passed. Functional neurology interventions may help reset these dysfunctional neurological programs, thereby restoring normal lactase production and alleviating symptoms of lactose intolerance [15].

The functional neurology intervention successfully eliminated the patient’s symptoms of lactose intolerance, such as frequent diarrhea, bloating, and severe gastrointestinal distress upon consuming dairy products. Despite these improvements, biochemical markers at higher time points (150 and 175 min) post-treatment remained similar to pre-treatment values, indicating persistent lactose malabsorption and highlighting the variability of the hydrogen measurements. This underscores the necessity for further research involving multiple tests or subjects to validate these preliminary findings. This discrepancy suggests that while functional neurology may effectively reset dysfunctional neurological programs and alleviate symptoms, the underlying biochemical response to lactose ingestion may remain unchanged. Then, the persistence of high hydrogen and methane levels at 150 and 175 min indicates that the biochemical processes underlying lactose malabsorption were not fully normalized. This observation aligns with the understanding that while functional neurology can modify neural pathways and alleviate symptoms, it may not directly impact the gut microbiota’s role in fermenting undigested lactose [16]. The sustained high levels of these gases suggest that the gut bacteria continued to ferment lactose, producing hydrogen and methane, even after the neurological symptoms were addressed. This points to the complex interplay between the nervous system and gut microbiota in lactose intolerance. The findings highlight the need for a comprehensive treatment approach that includes both neurological and dietary interventions to fully address the condition [17]. Despite significant symptomatic relief, the persistence of elevated biochemical markers suggests that functional neurology may not fully normalize lactose metabolism. This discrepancy highlights the complexity of lactose intolerance, which may involve both neurological and gut microbiota factors. The literature supports the idea that CNS trauma can affect digestive enzyme production [18,19,20]. Further studies are needed to explore this relationship and validate the findings of this case report.

This study has several limitations that should be acknowledged. Firstly, it is a single case report, which limits the generalizability of the findings. Further studies involving larger sample sizes are necessary to validate the results and determine the broader applicability of functional neurology interventions for lactose intolerance. Also, the breath test used for lactose intolerance is not very specific, which may limit the accuracy of the diagnosis and interpretation of results. Additionally, the study lacked a long-term follow-up to assess the sustainability of the symptomatic relief provided by the intervention. Future research should include longitudinal studies to monitor the persistence of treatment effects over time. Moreover, the biochemical tests indicated that while symptoms were alleviated, the underlying lactose malabsorption persisted, suggesting the need for a multifaceted treatment approach that also addresses gut microbiota.

Case reports play a crucial role in the medical literature, as they provide valuable insights into new or rare conditions, therapeutic effects, and potential adverse events, thus paving the way for further research and advancements in clinical practice [21,22,23,24]. The findings from this case report suggest that functional neurology can be a valuable complementary therapy for managing symptoms of lactose intolerance. However, it is crucial to recognize that the observed improvements might include placebo effects, and further research with larger sample sizes and multiple tests is essential to substantiate these findings. This approach may be particularly beneficial for patients who do not respond adequately to dietary modifications alone. Practitioners could consider integrating functional neurology techniques to address the neurological aspects of food intolerances, potentially improving patient outcomes. Additionally, this study highlights the importance of personalized treatment plans that consider both neurological and gastrointestinal factors.

In conclusion, this case report demonstrates that a single session of functional neurology can significantly alleviate the symptoms of lactose intolerance in the observed case. However, the persistence of elevated biochemical markers and the potential for placebo effects underscore the need for comprehensive treatment strategies and further research involving larger sample sizes and multiple tests. These findings underscore the need for a comprehensive treatment strategy that includes both neurological and dietary interventions. Further research with larger sample sizes and long-term follow-up is essential to fully understand the potential of functional neurology in treating lactose intolerance and to develop integrated treatment protocols that address both symptomatic relief and underlying biochemical imbalances.

## Figures and Tables

**Figure 1 life-14-00978-f001:**
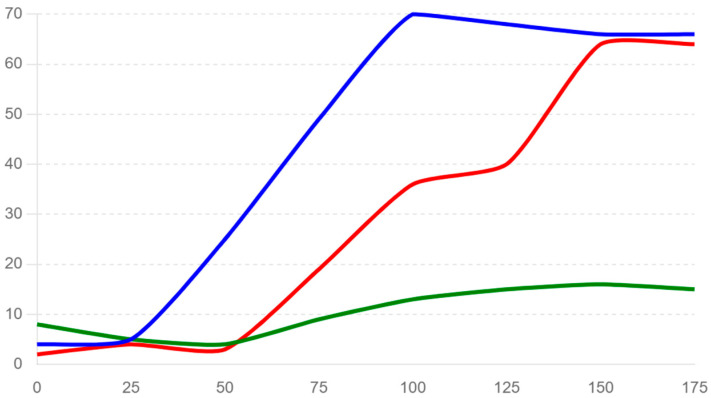
Changes in hydrogen and methane levels over time. X: gas levels (ppm); Y: time (minutes); The blue line represents the pre-treatment hydrogen levels, the green line represents the pre-treatment methane levels, and the red line represents the post-treatment combined hydrogen and methane levels.

**Table 1 life-14-00978-t001:** Hydrogen and Methane Levels Before and After Functional Neurology Treatment.

Time (Minutes)	Pre-Treatment Hydrogen (ppm)	Pre-Treatment Methane (ppm)	Post-Treatment Hydrogen + Methane (ppm)
0	4	8	2
25	5	5	4
50	25	4	3
75	49	9	19
100	70	13	36
125	68	15	40
150	66	16	64
175	66	15	64

## Data Availability

Data are contained within the article.

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
