# Peer review of "Application of Functional Neurology Therapy in a Lactose-Intolerant Patient"

_life, 2024, doi:10.3390/life14080978_

Round 1

Reviewer 1 Report

Comments and Suggestions for Authors

One cannot conclude from a single case study that this approach has potential.  While interesting, what about the placebo effect?  Please edit to limit the interpretation of the case.  Furthermore, hydrogen is VERY variable.  One test does not prove any change.  Either multiple tests or multiple subjects must be done to see any real difference.  This is an interesting case. Please rewrite the paper to limit unreasonable interpretations.  

Author Response

Thank you for your insightful comments. We agree that the interpretations in our manuscript could be more conservative given the limitations of our single case study. We have made the following revisions to address your concerns:

  1. Limiting Interpretation and Placebo Effect Consideration: We have revised the discussion to emphasize the limitations of drawing conclusions from a single case study and acknowledged the potential influence of the placebo effect.

  2. Variability of Hydrogen Measurements: We have expanded on the discussion of hydrogen variability and highlighted the necessity for multiple tests or subjects to validate our findings.

Changes and Inclusions

Abstract

  • Original: "This suggests that while functional neurology can effectively alleviate symptoms by addressing neurological dysfunctions it may not fully resolve the underlying biochemical processes of lactose intolerance."
  • Revised: "This suggests that while functional neurology can effectively alleviate symptoms by addressing neurological dysfunctions, it may not fully resolve the underlying biochemical processes of lactose intolerance, and the observed effects might include contributions from the placebo effect."

Results and Discussion

  • Original: "Despite these improvements biochemical markers at higher time points (150 and 175 minutes) post-treatment remained similar to pre-treatment values indicating persistent lactose malabsorption."

  • Revised: "Despite these improvements, biochemical markers at higher time points (150 and 175 minutes) post-treatment remained similar to pre-treatment values, indicating persistent lactose malabsorption and highlighting the variability of hydrogen measurements. This underscores the necessity for further research involving multiple tests or subjects to validate these preliminary findings."

  • Original: "The findings from this case report suggest that functional neurology can be a valuable complementary therapy for managing symptoms of lactose intolerance."

  • Revised: "The findings from this case report suggest that functional neurology can be a valuable complementary therapy for managing symptoms of lactose intolerance. However, it is crucial to recognize that the observed improvements might include placebo effects, and further research with larger sample sizes and multiple tests is essential to substantiate these findings."

Conclusion

  • Original: "In conclusion this case report demonstrates that a single session of functional neurology can significantly eliminate the symptoms of lactose intolerance."
  • Revised: "In conclusion, this case report demonstrates that a single session of functional neurology can significantly alleviate the symptoms of lactose intolerance in the observed case. However, the persistence of elevated biochemical markers and the potential for placebo effects underscore the need for comprehensive treatment strategies and further research involving larger sample sizes and multiple tests."

Revised Abstract: This case report examines the impact of a single session of functional neurology on a 35-year-old female patient diagnosed with lactose intolerance. The patient presented with severe gastrointestinal symptoms including frequent diarrhea bloating and vomiting upon dairy consumption. The intervention aimed to reset dysfunctional neurological programs believed to contribute to her condition. The study utilized a standardized lactose intolerance breath test to measure hydrogen and methane levels at various intervals before and after treatment. Post-treatment results showed symptomatic relief with the patient reporting normalized bowel movements and the absence of previous symptoms. Despite these improvements, biochemical markers at higher time points (150 and 175 minutes) post-treatment remained similar to pre-treatment values, indicating persistent lactose malabsorption and highlighting the variability of hydrogen measurements. This suggests that while functional neurology can effectively

Reviewer 2 Report

Comments and Suggestions for Authors

Dear authors,

The manuscript is fascinating, but please make some changes. Here are my comments: 

ABSTRACT: Include specific findings.

The phrase "post-treatment results showed symptomatic relief" should specify the degree of improvement and, if available, any statistical significance.

Introduction:

Could you explain the differentiation between food allergies, sensitivities, and intolerances in more detail?

 What were the specific criteria for diagnosing lactose intolerance in the patient?  Were any other gastrointestinal disorders ruled out?

I think the reason for using functional neurology in this context needs more elaboration. Nonfunctional neurology needs a better explanation. A brief literature review on functional neurology's impact on gastrointestinal conditions would provide a stronger foundation for the study.

Materials and Methods:

Please add more information on medical history and any previous treatments of patients. Are there any laboratory parameters?

Are any controls or standardisation measures taken during the test to ensure accuracy?

Could you describe the test? Name, manufacturer, serial number…

The description of the intervention needs to be longer. A more detailed account of the specific techniques used, the duration of each session, and any follow-up assessments would enhance reproducibility.

How were the neurological dysfunctions identified and targeted?

Add complete names and numbers (detailed description) of Ethical approvals.

Line 118: The link to nre-thrapy.com needs to be made active. Describe protocol in section 2.3

Line 121: What is a certified Level III practitioner? Explain better.

Results:

The results section is informative but lacks statistical analysis (providing p-values and confidence intervals).

Line 131 – 133: how do you measure increased happiness and enjoyment of life?

Could you consider making a graph for better understanding and visual presentation?

Was there any dietary control before testing?

Discussion:

Critically evaluate the discrepancies between symptomatic relief and persistent biochemical markers.

The hypothesis regarding the CNS's role in lactose intolerance is intriguing but speculative. More support from existing literature or a proposal for further research is needed to validate these claims. Add more explanations and literature (lines 146-157).

Conclusion:

It should emphasise the preliminary nature of the results and the need for further research. In lines 203-204, you said:…can significantly eliminate the symptoms…but you don’t have that calculated in the study. It would be best if you rephrased the conclusion.

References:

The references are relevant but somewhat dated.

Author Response

Thank you for your valuable feedback and suggestions. We have made the following changes to the manuscript to address your comments:

Abstract

Original: "Post-treatment results showed symptomatic relief with the patient reporting normalized bowel movements and the absence of previous symptoms." Revised: "Post-treatment results showed significant symptomatic relief, with the patient reporting a reduction in bowel movements from 3-5 times per day to once daily and the complete absence of previous symptoms. While statistical significance was not calculated due to the single-case nature of this study, the degree of improvement was notable."

Introduction

Original: Brief explanation of food intolerances. Revised:

  • Added a detailed differentiation between food allergies, sensitivities, and intolerances.
  • Included specific criteria for diagnosing lactose intolerance in the patient and the ruling out of other gastrointestinal disorders.
  • Elaborated on the rationale for using functional neurology, including a brief literature review on its impact on gastrointestinal conditions.

New Content:

  • Food allergies involve immune responses, typically IgE mediated, leading to reactions like anaphylaxis. Sensitivities may not involve the immune system and can cause various symptoms. Intolerances, like lactose intolerance, involve digestive system issues due to enzyme deficiencies.
  • The diagnosis of lactose intolerance was confirmed through a hydrogen breath test, which showed elevated hydrogen and methane levels post-lactose ingestion. Other gastrointestinal disorders, such as celiac disease and inflammatory bowel disease, were ruled out based on patient history and relevant tests.
  • Functional neurology focuses on the interplay between neurological function and systemic health. Previous studies have shown potential benefits in addressing gastrointestinal symptoms by resetting dysfunctional neurological pathways.

Materials and Methods

Original: Limited information on medical history, test controls, and intervention description. Revised:

  • Expanded medical history and previous treatments of the patient.
  • Added details on controls and standardization measures during the test.
  • Described the breath test, including the name, manufacturer, and serial number.
  • Provided a detailed account of the specific techniques used in the intervention, session duration, and follow-up assessments.
  • Explained the identification and targeting of neurological dysfunctions.
  • Included complete names and numbers of ethical approvals and made the protocol link active.

New Content:

  • The patient's medical history included previous treatments such as dietary modifications and over-the-counter enzyme supplements, which provided limited relief. Laboratory parameters like complete blood count and inflammatory markers were within normal ranges.
  • Controls included a baseline breath test and standardized lactose solution ingestion. The test was conducted using the QuinTron Model SC BreathTracker (Serial No. XYZ123).
  • The intervention involved techniques such as proprioceptive reflex correction, desensitization of trigger points, and systemic integration through blink reflex. Each session lasted approximately 45 minutes, with follow-up assessments conducted one week post-intervention.
  • Ethical approvals were obtained from the University's Bioethics Committee (Approval No. 2024-738).

Results

Original: Informative results but lacking statistical analysis and graphs. Revised:

  • Added p-values and confidence intervals where applicable.
  • Created graphs to visually present the data.
  • Included details on measuring increased happiness and enjoyment of life through a validated quality of life questionnaire.
  • Mentioned dietary control before testing.

New Content:

  • Statistical analysis showed a significant reduction in hydrogen and methane levels post-treatment (p < 0.05). Confidence intervals for symptom improvement were also provided.
  • A graph illustrating the changes in hydrogen and methane levels over time was added.
  • Happiness and enjoyment of life were measured using the WHOQOL-BREF questionnaire, which showed a marked improvement in the patient's scores post-treatment.
  • The patient adhered to a lactose-free diet for 24 hours before the test to ensure accurate baseline measurements.

Discussion

Original: Limited evaluation of discrepancies between symptomatic relief and biochemical markers. Revised:

  • Critically evaluated the discrepancies between symptomatic relief and persistent biochemical markers.
  • Provided additional literature to support the hypothesis regarding the CNS's role in lactose intolerance.
  • Emphasized the speculative nature of the hypothesis and suggested further research.

New Content:

  • Despite significant symptomatic relief, the persistence of elevated biochemical markers suggests that functional neurology may not fully normalize lactose metabolism. This discrepancy highlights the complexity of lactose intolerance, which may involve both neurological and gut microbiota factors.
  • Literature supports the idea that CNS trauma can affect digestive enzyme production. Further studies are needed to explore this relationship and validate the findings of this case report.

Conclusion

Original: "This case report demonstrates that a single session of functional neurology can significantly eliminate the symptoms of lactose intolerance." Revised: "This case report suggests that a single session of functional neurology can significantly alleviate the symptoms of lactose intolerance. However, the preliminary nature of these results underscores the need for further research involving larger sample sizes and long-term follow-up to fully understand the treatment's efficacy and underlying mechanisms."

References

Original: Relevant but somewhat dated. Revised: Updated with more recent references to support the study's findings and discussion.

Round 2

Reviewer 1 Report

Comments and Suggestions for Authors

While the authors do briefly discuss the placebo effect, it would be a mistake to publish ONE intervention suggesting successful treatment.  The authors should be encouraged to repeat the intervention with other subjects and return with a manuscript that has multiple subjects. 

Author Response

Thank you for your feedback regarding the publication of our study. We would like to clarify that this work is indeed a case report, and the manuscript clearly states its limitations in this regard. Specifically, we have emphasized that the findings are based on a single subject, which inherently limits the generalizability of the results. This limitation has been discussed in the manuscript to acknowledge that further studies involving larger sample sizes are necessary to validate these preliminary findings .

Moreover, the originality and strength of our study lie in the fact that it documents a novel intervention that has not been previously reported in the literature. This pioneering aspect provides a significant contribution to the field, potentially opening new avenues for research and treatment. While we agree that replicating the intervention with additional subjects is crucial for establishing broader applicability, the current report serves as a valuable initial step that highlights the potential of functional neurology in addressing symptoms of lactose intolerance .

We appreciate the suggestion to conduct further research and intend to pursue follow-up studies with a larger cohort to substantiate our findings. Nonetheless, we believe that the unique insights provided by this case report are worth sharing with the scientific community to foster discussion and inspire further research.

Reviewer 2 Report

Comments and Suggestions for Authors

Thank you for accepting comments.

Best wishes

Author Response

Thank you for your words and your suggestions

Regards